# A Diversity Model Based on Dimension Entropy and Its Application to Swarm Intelligence Algorithm

**DOI:** 10.3390/e23040397

**Published:** 2021-03-27

**Authors:** Hongwei Kang, Fengfan Bei, Yong Shen, Xingping Sun, Qingyi Chen

**Affiliations:** School of Software, Yunnan University, Kunming 650000, China; hwkang@ynu.edu.cn (H.K.); ynubei@mail.ynu.edu.cn (F.B.); sunxp@ynu.edu.cn (X.S.); devas9@ynu.edu.cn (Q.C.)

**Keywords:** swarm intelligence algorithm, diversity model, dimension entropy

## Abstract

The swarm intelligence algorithm has become an important method to solve optimization problems because of its excellent self-organization, self-adaptation, and self-learning characteristics. However, when a traditional swarm intelligence algorithm faces high and complex multi-peak problems, population diversity is quickly lost, which leads to the premature convergence of the algorithm. In order to solve this problem, dimension entropy is proposed as a measure of population diversity, and a diversity control mechanism is proposed to guide the updating of the swarm intelligence algorithm. It maintains the diversity of the algorithm in the early stage and ensures the convergence of the algorithm in the later stage. Experimental results show that the performance of the improved algorithm is better than that of the original algorithm.

## 1. Introduction

### 1.1. Optimization Problems and Swarm Intelligence Algorithms

Optimization problems have a long history. They involve the need to determine specific performance requirements for a certain problem if there are multiple alternative solutions, and to select one of them to maximize or minimize the determined performance requirement index [1]. In real life, optimization problems exist widely in engineering design [2], image segmentation [3], power systems [4], and other fields. In order to better solve optimization problems, evolutionary algorithms simulating the process and mechanism of biological evolution and swarm intelligence algorithms simulating the foraging mechanism of biological populations have gradually become a research hotspot in recent years.

Swarm intelligence refers to the behavior of group cooperation and collective intelligence presented by a group composed of many simple individuals in nature [5]. Swarm intelligence is a kind of group-based computing method with self-organization, self-adaptation, and self-learning characteristics, which is put forward by referring to and utilizing various mechanisms of natural phenomena or organisms in nature. After years of development, a large number of swarm intelligence optimization algorithms have been born, among which the classic swarm intelligence optimization algorithms include the artificial bee colony algorithm [6], the ant colony algorithm [7], and the particle swarm optimization algorithm [8].

### 1.2. An Overview of the Diversity of Swarm Intelligence Algorithms

A major problem in swarm intelligence algorithms is premature convergence [9,10,11,12]; i.e., algorithms lose diversity prematurely. The root cause of premature convergence is the imbalance between local exploration and global development [13]. Too much local exploration leads to premature convergence of the algorithm into a local optimum, while too much global development causes the algorithm to lack precision and become difficult to converge [14].

To some extent, exploration and development can be seen as a pair of contradictory concepts, and the increase of one will inevitably reduce the other [15]. Population size, search strategy, and restart strategy are all effective methods to control exploration and development, but how to balance them scientifically is the key for a swarm intelligence algorithm to achieve excellent results.

Evaluating the diversity of algorithms can fully detect the exploration and development of algorithms. As an important index of the swarm intelligence algorithm [16], diversity measures the richness of particle position, cognition, direction, and other properties in the swarm intelligence algorithm. Existing studies on diversity and entropy include: Folino et al. proposes a method to evaluate swarm intelligence algorithm using entropy value [17]; González et al. proposes a natural inspired strategy for optimization [18]; Muhammad et al. proposed a design with entropy evolution of optimal power flow problem [19]. Da et al. proposed a simplex crossover evolutionary algorithm aiming at genetic diversity [20].

According to the benchmark and representation method of the diversity model, there can be many diversity models. Therefore, the scope of this paper should be determined first. The diversity studied in this paper is a measure related to the position of individuals in a population. We believe that a measure model that can measure the true diversity of a population should have the following properties, besides being true and effective:(1)It is robust to parameters such as population size and problem dimensions.(2)It has repeatability for different populations.(3)It can give feedback directly to changes in the population.

For this reason, we have designed a new diversity model. The model calculates population entropy based on particle position, which is named “dimensional entropy”. Compared with other methods, dimensional entropy can clearly and intuitively define the diversity of a population and thus control the iteration of the algorithm. This paper is organized as follows: This section introduces some basic knowledge about the swarm intelligence algorithm and its diversity; in the second section, we describe a variety of diversity models and discuss the dimensional entropy model. Section 3 shows the method of updating the swarm intelligence algorithm guided by the dimensional entropy model; Section 4 shows the results; Section 5 presents our conclusions.

## 2. Diversity Model Based On Dimension Entropy

### 2.1. General Concept

Generally speaking, diversity [21,22] can be defined as the degree of individual heterogeneity between populations [23]. In swarm intelligence algorithms, there are many kinds of diversity evaluation models, which can be divided into two categories. The first type is the measurement based on the distance between particles [24,25]. This distance can be based on a central particle [26], the maximum distance between two particles in space [27,28], or the average distance between particles [29]. Euclidean distance is a common calculation method, because Euclidean space extends two-dimensional or three-dimensional space to any dimension, and at higher dimensions, populations exist and are defined in Euclidean space.

The second type is based on the entropy measure. Entropy is a concept in thermodynamics. In 1948, Shannon put forward the concept [30]. The idea is that calculating the probability of several independent random discrete events can measure how much of an information system is uncertain. When only one event occurs in an information system, information entropy is the minimum; when the probability of occurrence of multiple discrete events is the same, the information entropy is the maximum.

When applied to the swarm intelligence algorithm, because the population itself is a continuous concept, the first step is to segment the population, i.e., to discretize it. Thus, each segmented interval is abstracted as a random event, and the ratio of the number of particles contained in each interval to the total population is the probability of this “random event” occurring. Therefore, how to discretize and which standard to discretize become a key, difficult point in applying the entropy standard in the swarm intelligence algorithm. At the same time, the number of intervals divided by discretization has a direct impact on the estimation of diversity. When the ethnic scale is too small, the entropy method cannot be used to divide enough intervals. However, in the case of high dimensions, it is difficult to consider the situation of all dimensions when dividing the interval.

In addition, the combination of entropy values between all the particles must be considered. For example, Gouvea Jr. and Araujo used a representative particle to represent population diversity [28]; i.e., they used the individual characteristics of a particle to represent population diversity. They mentioned that the chosen particle has to be important. Collins and Jefferson used average entropy values to determine population diversity [31].

### 2.2. Research Status of Diversity

To save space, a summary of the symbols used in this section is first given in Table 1 below.

The most basic range-based diversity measure only considers the diameter of the population, i.e., the distance between the two farthest particles in the population [30]. The formula is shown in Equation (1):(1)$$D_{d}=max\left(\sqrt{\overset{n}{\underset{k=1}{\sum }}{\left(x_{i,k}-x_{j,k}\right)}^{2}}\right)$$

The second method [32] can be obtained by changing the farthest distance to the radius of the population and calculating the distance between the farthest particle and the average position of the population. The formula is shown in Equation (2):(2)$$D_{r}=max\left(\sqrt{\overset{n}{\underset{k=1}{\sum }}{\left(x_{i,k}-\bar{x_{k}}\right)}^{2}}\right)$$

There are other extended methods of this method, such as calculating the average radius, which will not be detailed here.

The third method was proposed by Olorunda and Engelbrecht. The idea is to consider the mean value of the mean distance around the population particles [27], as shown in Equation (3):(3)$$D_{all}=\frac{1}{N}\overset{N}{\underset{i=1}{\sum }}(\frac{1}{N}\overset{N}{\underset{j=1}{\sum }}\sqrt{\overset{n}{\underset{k=1}{\sum }}{\left(x_{i,k}-x_{j,k}\right)}^{2}})$$

This method has a huge calculation cost. In order to save the calculation cost, Wineberg and Opacher proposed a calculation method named “true diversity” [33], which represents the mean standard deviation of each particle in the population, as shown in Equation (4):(4)$$D_{td}=\frac{1}{n}\sqrt{\overset{n}{\underset{k=1}{\sum }}\left(\bar{x_{k}^{2}}-{\left(\bar{x_{k}}\right)}^{2}\right)}$$ where: $$\bar{x_{k}^{2}}=\frac{1}{N}\overset{N}{\underset{i=1}{\sum }}x_{i,k}^{2}$$


The last diversity based on distance measure compared in this paper was proposed by Herrera and Lozano [26]. This diversity measure requires pre-determination of the most suitable particle in the population, because it uses this particle as a reference to measure the distance with other particles, as shown in Equation (5):(5)$$D_{e}=\frac{\bar{d}-d_{min}}{d_{max}-d_{min}}$$
where: $$\begin{array}{c} \bar{d}=\frac{1}{N}\overset{N}{\underset{i=1}{\sum }}\sqrt{\overset{n}{\underset{k=1}{\sum }}{\left(x_{i,k}-x_{best,k}\right)}^{2}}\\ d_{max}=\;\underset{i\in \left(1,2,\ldots ,N\right)}{\max}\left(\sqrt{\overset{n}{\underset{k=1}{\sum }}{\left(x_{i,k}-x_{best,k}\right)}^{2}}\right)\\ d_{min}=\;\underset{i\in \left(1,2,\ldots ,N\right),i\neq best}{\min}\left(\sqrt{\overset{n}{\underset{k=1}{\sum }}{\left(x_{i,k}-x_{best,k}\right)}^{2}}\right) \end{array}$$

In terms of the entropy value measurement, Shannon’s entropy is the most basic measurement method [34], which measures the disorder degree of the population [35]. Its entropy definition is shown in Equation (6):(6)$$E=-\overset{M}{\underset{m=1}{\sum }}p_{m}log\left(p_{m}\right)$$

If we want to measure the entropy value of the swarm intelligence algorithm, it is important to establish a discrete measurement model. Chen et al. put forward an idea based on entropy calculation methods of fitness [36]. The idea is to investigate the historically best position. The interval is defined as the current particle fitness range, which can be divided by the same number of particles of M between communities, so as to calculate the distribution of the current fitness [37]. The formula is shown in (7):(7)$$E=-\overset{M}{\underset{m=1}{\sum }}p_{m}log\left(p_{m}\right)$$
where
$$p_{m}=\frac{k_{3}}{M}$$
where
$k_{m}$. represents the number of particles contained in interval m.

Wang and Lei put forward the intuitionistic fuzzy population entropy as a measure of diversity [38,39]. In this method, the PBest position of each generation is selected as the aggregation point, the distance from particle j to the aggregation point i is $D_{ji}$, and the scope radius of each aggregation point is $R_{i}=K*\;Max\;\left(D_{j}\right)$, where K is a random number between $\left[0,0.5\right]$. If the distance $D_{ji}$ from particle j to the aggregation point i is less than the scope radius $R_{i}$ of the aggregation point i, then particle  j is considered to belong to the aggregation point i and the counter $T_{i}$ is increased by 1. If particle j does not belong to any of the scopes, then particle j is considered as an “lone point” and the global counter $T_{x}$ is added 1. The formula is shown in (8):(8)$$E=\frac{1}{M}\overset{M}{\underset{i=1}{\sum }}\frac{\min\left(\mu_{ji},\gamma_{ji}\right)+\pi_{ji}}{\max\left(\mu_{ji},\gamma_{ji}\right)+\pi_{ji}}$$
where:$$\begin{array}{c} \mu_{ji}=\frac{T_{ji}}{M}\\ \pi_{ji}=\frac{T_{x}}{M}\\ \gamma_{ji}=1-\mu_{ji}-\pi_{ji} \end{array}$$

The intuitionistic fuzzy population entropy reflects the aggregation degree of particles in the algorithm solution process.

### 2.3. Some Fundamental Flaws in Current Metrics

First of all, population diameter is not an ideal diversity measurement method, because this method only considers the two farthest particles and ignores the distribution of the remaining particles, which cannot explain the true diversity of a population.

A similar deficiency exists in population radius, where the diversity is based on the position of the particles furthest from the population center. When this indicator describes a fully diversified population, the value is close to 0.5. As the value approaches 1, it describes a population that is mostly clustered around one corner and has an outlier near the other corner. This diversity is also wrong.

The diversity based on the distance measure needs to determine a reference particle in advance, but in fact, a suitable reference particle is difficult to choose. Meanwhile, for a linearly shrinking population, its diversity will remain unchanged because of the simultaneous contraction of the numerator and denominator, and the real decreased diversity cannot be described.

It is difficult to determine an appropriate segmentation standard using the diversity measure based on entropy. The entropy calculation based on the fitness of particles has the same fitness and the same interval. It may work with a single peak function, but in the face of two likely multimodal functions for different peak particles with the same fitness, a simple classification by fitness is rigorous.

Intuitionistic fuzzy population entropy considers the scope division of particles in space, but the difficulty of this division increases greatly with the increase of dimensions. In high dimensions, a particle is likely to belong to different scopes in different dimensions, so it is difficult to clearly divide. Finally, as we will see later, most of the metrics fail to deal with population dynamics.

### 2.4. Diversity Model Based on Dimension Entropy

Xu and Cui [40] confirmed that the swarm intelligence algorithm in the iteration is relatively independent of each dimension in the process of change. Inspired by this, this paper puts forward dimension entropy to measure the diversity of the swarm intelligence algorithm. Unlike other entropy value methods, we abandoned the concept of space. We have an independent view of the entropy value of each dimension. Dimension entropy is put forward. We provide relevant definitions:

*Dimension interval.* The maximum and minimum values of each dimension were selected as the upper and lower limits and divided into M intervals (M is the total number of particles) on an average basis. Each dimension was divided into intervals independently without interference.

*Dimensional entropy.* In each dimension, the number of particles falling into each dimension interval is counted independently, and the dimension entropy is calculated, as shown in Equation (9):(9)$$E_{dim}=-\frac{1}{n}\overset{n}{\underset{k=1}{\sum }}\left(\overset{M}{\underset{m=1}{\sum }}p_{m,k}log\left(p_{m,k}\right)\right)$$
where:$$p_{m,k}=\frac{k_{m,k}}{M}$$
where $k_{m,k}$ is the number of particles contained in the m interval of dimension  k.

In the worst-case scenario, the population is trapped in the same interval in all dimensions; that is, the population completely converges, and the dimensional entropy is then 0. The larger the dimensional entropy, the greater the diversity of the population.

Compared with the previous entropy method, our dimension entropy method perfectly overcomes the disadvantages of the usual entropy methods. With a low population size, even if some dimensions cannot be effectively divided, the entropy can be calculated as long as one of the dimensions can complete the interval division. At the same time, the independent thinking among the dimensions also enables us to face the challenges of higher dimensions simply and intuitively.

### 2.5. A Comparative Study

In order to compare the differences between different diversities, the various methods should be normalized first so as to evaluate on the same standard. In this study, the maximum value was used as the normalization factor. After normalization, the value range of each model was 0~1.

In addition to the dimension of entropy, we selected the diversity model to participate in six kinds of contrast: the maximum distance [32], the radius of the population [32], the individual average distance [27], and the average standard deviation [33] , two entropy value methods: the fitness [36] and the intuitionistic fuzzy entropy [38]. These are marked, respectively, as $D_{dp}$, $D_{rp}$, $D_{all}$, $D_{td}$, $E_{fit}$, $E_{fuzzy}$.

#### 2.5.1. Population Expansion Experiment

The practical application scenario of the swarm intelligence algorithm is complex, which will inevitably face population expansion or decline. A good diversity model should have a correct and timely response to changes of population size.

In order to compare the robustness of various diversity models of population size, we designed the following experiments.

First, we designed a complete population of 20 particles with a population range of [−100, 100]. This population contains two dimensions and has the following characteristics:(1)All particles in the population are uniformly distributed in all dimensions.(2)No two particles are the same.

The complete population is shown in Figure 1.

We randomly selected the composition of the initial population and the population expansion after the simulation operation from a population of 20 particles. The initial population consists of 10 particles. We randomly selected 1 out of 10 particles and joined the initial population. After 10 iterations, the entire population was produced, as shown in Figure 1. The process of population change is in Figure 2.

Since there is a significant difference between each particle in the complete population, it can be seen that, in the population expansion process in the figure above, each new particle is different from the old particle, so the diversity of the population must continue to increase in this process.

We used the above several diversity models to evaluate the population diversity in this process. The entropy value method is based on the population fitness. We used a simple Sphere function ($f\left(x\right)=\overset{k}{\underset{i=1}{\sum }}{\left(x_{i}\right)}^{2}$) to calculate the entropy. Because the fuzzy population entropy needs to use the PBest value of the past dynasties as the aggregation point in the calculation process, it was not used in this experiment.

The normalized results of various models are shown in Figure 3.

According to the above figure, the population diversity of $D_{dp}$ increased only in the fourth iteration, because the particles added in the fourth iteration happened to be outside the existing population, while in the remaining iterations, the particles were added inside the population, which means that the simple method to calculate the population diameter could not judge the changes within the population at all.

Although $D_{rp}$ shows an overall upward trend, its value is always at a high level, which means that, if the new particle is not a significant outlier, then $D_{rp}$ cannot make a clear response to it. Throughout all 10 expansion processes, the diversity value of $D_{rp}$ decreased four times, because, while the new particles did not expand the maximum radius of the population, the enlarged denominator decreased the value of $D_{rp}$.

$D_{all}$ has a similar problem. If the new particles do not increase the average distance between the particles, $D_{all}$ will not reflect the population change correctly.

$D_{td}$ kept a downward trend during the iteration, because the increase in particles resulted in a decrease in the standard deviation, which proved that $D_{td}$ could not be used to describe a changing population.

$E_{fit}$ overall performs better, but still showed a diversity, which reduced the phenomenon. This is because the eighth iteration was added on the right, and existing particles around the zero point were symmetrical. The two particles were completely different; however, in our set, the fitness function is similar. The visible fitness does not have the unique attributes of a particle. How unique and accurate properties are chosen as division standards is a key part of the entropy value method.

Finally, among all the methods, only $E_{dim}$ keeps an upward trend at all times, and the upward trend is visible in each iteration, which is sufficient to prove that our proposed method can accurately observe and describe a changing population.

#### 2.5.2. Dimensional Change Experiment

The swarm intelligence algorithm also faces a complex dimension problem. We hope that the diversity of an ideal method is robust for different dimensions; namely, if the population itself is fully diversified, then regardless of dimensions, its diversity values should be stable; otherwise, it would be difficult to assess or apply in the swarm intelligence algorithm.

To this end, we designed an experiment. First, we built a completely differentiated network with 1 dimension and 100 particles, and then continuously expanded the dimensions on this basis, ensuring that each expanded dimension was also completely differentiated. This step was repeated until it reached 30 dimensions. In this process, we used the above diversity model to compare the values of each model for different dimensions, and the results are shown in Figure 4.

As can be seen in Figure 4, with the increase in dimensions, all distance diversity models show an increasing trend of diversity. This is because, with the increase in dimensions, the spatial span calculated by the diversity model based on spatial distance also increases proportionally, which is almost an inevitable defect of the distance diversity model. In contrast, the entropy method has better resistance to dimensional changes, and compared with $E_{fit}$, the method we defined is more accurate and stable.

#### 2.5.3. Practical Problem Testing

The purpose of this experiment is to test the performance of each model under the actual test function environment. First of all, we are given a specific Rastrigin function test. The test function exists, and there is only one global optimal value at the zero point; therefore, while the algorithm is run, almost all particles, according to certain trends, will move toward the optimal point 0 mobile. As the iteration times begin to approach zero, and when the late algorithm flocks to near zero, all particles moving toward the zero distance sum partly reflect the diversity of the population under the test function. The formula of the Rastrigin test function is shown in Equation (10):(10)$$F\left(x\right)=\overset{k}{\underset{i=1}{\sum }}\left(x_{i}^{2}-10\times \cos\left(2\times \pi \times x_{i}\right)+10\right)$$

The standard PSO algorithm was used to carry out the experiment. The experimental dimension was 5 dimensions, there were 50,000 fitness evaluations, the population size was set as 100, and the diversity models involved in the experiment were $D_{dp}$, $D_{rp}$, $D_{all}$, $D_{td}$, $E_{fit}$, $E_{fuzzy}$, and $E_{dim}$. The PSO setup is described in the next section.

The Spearman correlation coefficient is a nonparametric index to measure the dependence of two variables. It uses monotone equations to evaluate the correlation between two statistical variables. When the two variables are completely monotone and positively correlated, the Spearman correlation coefficient is 1; when the two variables are completely monotone and negatively correlated, the Spearman correlation coefficient is −1. That is to say, the closer the Spearman correlation coefficient is to 1, the closer the two variables are.

The diversity model mentioned above was used to compare the distance changes of all particles to the point 0. After each iteration, we calculate and record the sum of the distances of the all particles from zero point and the values of the above diversity model. At the end of the algorithm, we selected the values of each diversity model in each iteration and compared them with the sum of the distances of the all particles from zero point to calculate Spearman correlation coefficient. The experiment was repeated ten times, the results are shown in Table 2.

In the case of fixed population size and dimension, $D_{all}$ performs best, followed by $E_{dim}$. Among all entropy methods, $E_{dim}$ has a significant advantage.

## 3. Swarm Intelligence Algorithm Control Method Based on Dimension Entropy

### 3.1. Introduction to Swarm Intelligence Algorithms

Firstly, the comparison algorithm used in this study is briefly introduced.

PSO is derived from an analogy of bird flight [41]. In PSO, each particle flies in a D-dimensional solution space by learning its own experience and the experience of its neighbor. The position and velocity of each particle are expressed by $X_{i}=\left[x_{i}^{1},x_{i}^{2},\ldots ,x_{i}^{D}\right]$ and $V_{i}=\left[v_{i}^{1},v_{i}^{2},\ldots ,v_{i}^{D}\right]$, respectively. $X_{i}$ represents the current position of the i particle, and it also represents a possible solution of the optimization problem. $V_{i}$ represents the current velocity of the i particle, which determines the direction and step size of the particle’s movement. Each particle learns from its own and global historical best positions, represented by $pbest_{i}=\left[pbest_{i}^{1},pbest_{i}^{2},\ldots ,pbest_{i}^{D}\right]$ and $gbest_{i}=\left[gbest_{i}^{1},gbest_{i}^{2},\ldots ,gbest_{i}^{D}\right]$, respectively. The position and velocity of each particle are dynamically adjusted according to equations (11) and (12):(11)$$v_{i}^{d}\left(t+1\right)=\omega \times v_{i}^{d}\left(t\right)+c_{1}\times rand1_{i}^{d}\times \left(pbest_{i}^{d}\left(t\right)-x_{i}^{d}\left(t\right)\right)+c_{2}\times rand2_{i}^{d}\times \left(gbest_{i}^{d}\left(t\right)-x_{i}^{d}\left(t\right)\right)$$
(12)$$x_{i}^{d}\left(t+1\right)=x_{i}^{d}\left(t\right)+v_{i}^{d}\left(t+1\right)$$
where ω is the inertial weight, $c_{1}$ and $c_{2}$ are the acceleration factors, and $rand1_{i}^{d}$ and $rand2_{i}^{d}$ are two uniformly distributed random numbers in the range of [0,1].

Bare-bones PSO eliminates the velocity attribute of particles, and the position update formula of particles is obtained by random sampling in accordance with Gaussian distribution, which can be described in a mathematical language as follows:

The particle is searched randomly in the M-dimensional space, the position of particle i is $X_{i}=\left[x_{i}^{1},x_{i}^{2},\ldots ,x_{i}^{D}\right]$, and the optimal flight position of particle i is $pbest_{i}=\left[pbest_{i}^{1},pbest_{i}^{2},\ldots ,pbest_{i}^{D}\right]$. PBest is the global optimal position of the particle, and the symbol N(0,1) is the standard Gaussian distribution. The position update formula of particle i is shown in Equation (13):(13)$$\begin{array}{c} X_{i}\left(t+1\right)=\mu_{i}\left(t\right)+N\left(0,1\right)\sigma_{i}\left(t\right)\\ \mu_{i}\left(t\right)=0.5\left(pbest\left(t\right)+gbest\left(t\right)\right)\\ \sigma_{i}\left(t\right)=\left|\mathrm{pbest}\left(t\right)-\mathrm{gbest}\left(t\right)\right| \end{array}$$

### 3.2. The Swarm Intelligence Algorithm Control Method Based on Dimension Entropy

The disadvantage of traditional swarm intelligence algorithms is the premature loss of diversity. We want to come up with a mechanism: control the diversity to decrease slowly, maintain higher diversity in the early stage, and reduce the diversity in the late stage to promote convergence. So we propose a method to calculate and control population diversity according to dimensional entropy. Firstly, the following definitions are proposed:

*Redundant particle*: Divide M (M is the population size) intervals on average in each dimension $d\in \left[1,2,\ldots ,n\right]$, count the number of particles in each interval. If a particle $i\in \left[1,2,\ldots M\right]$ is in the largest set of dimensions d, let count $k_{i}$+1, the particle with the largest count in a population is redundant particle.

It should be noted that redundant particles are not necessarily repeated particles or redundant particles, but according to the definition of dimensional entropy, we know that deleting redundant particles will inevitably increase dimensional entropy, while copying redundant particles will inevitably reduce dimensional entropy.

Based on this, we propose a strategy to control the diversity. When the diversity is too large, we copy a redundant particle and delete a particle with the worst fitness to reduce the diversity. On the contrary, when the diversity is too low, a redundant particle is deleted and a new random particle is added to achieve the purpose of increasing the diversity. The relevant algorithm pseudocode is shown in Algorithm 1:
**Algorithm 1 Diversity control** Input:
 $E_{dim}$: The dimensional entropy of the population in this iteration; $E_{standard}$: The expected entropy of this iteration; Coord: The population
 Output:
 The new population
 1 if $E_{dim}$< $E_{standard}$
 2  delete a redundant particle;
 3  add a new particle;
 4 else if $E_{dim}$> $E_{standard}$
 5  delete a worst particle;
 6  copy a redundant particle;
 7 end

We want to set a standard for diversity to go down on a trajectory, $E_{standard}$ is the entropy value we expect to achieve in each iteration. This value is determined by the base curve.

We have experimented with four different kinds of base curves: straight line, convex curve, concave curve and broken line. These curves are shown in Figure 5.

In order to compare the differences between different convergence criteria, we set a simple test function set for experiment. This test function set contains 7 basic test functions, the information is shown in Table 3.

In the above test function, taking the linear diversity reduction standard as shown in Figure 5 as an example, the PSO algorithm uses the strategy of Algorithm 1, and its diversity changes are shown in Figure 6.

According to the diversity change curve, the diversity of the original algorithm will be reduced to the lowest value after about 50 iterations, and then the algorithm will stagnate. This phenomenon of premature loss of diversity is one of the reasons why swarm intelligence algorithm cannot achieve better results.

After using our strategy, diversity slowly declines on a curve that we have defined. In the early stage of the algorithm, there is still a phenomenon of a sharp decrease in the diversity. This is due to the rapid aggregation of particles at the beginning of the algorithm. In the middle and late stages of the algorithm, our method can achieve precise control.

We take these four convergent curves as input to the algorithm, PSO algorithm was used as the experimental algorithm, the experimental dimension is 10 dimensions. The experiment was repeated for 30 times. Compare the difference of optimization performance and population diversity brought by different curves on different functions. Advantageous test function results are marked in bold. The result is shown in Table 4.

According to the table, first of all, the concave curve has the highest average diversity, followed by the straight line, followed by the broken line, and the concave curve has the lowest average diversity.

In terms of the optimization results, the convex curve has higher pre - and mid-term diversity, which guarantees its wider development capability in the pre - and mid-term. This ability enables it to better discover hidden global optimal values, which may be the reason for its excellent performance in multi-modal functions. In the case of unimodal function, appropriately accelerating the convergence can enhance the exploration ability of the algorithm, so the linear curve with slightly faster multiplicity convergence can achieve better results. And for any function, a rapid loss of diversity is not a good strategy, as concave curves demonstrate.

## 4. Experiment and Discussion

In this section, we apply the improved strategy to the three algorithms mentioned in the previous chapter. CEC17 is selected as the test function, which is a test function set containing 29 functions. Among them, $f_{1}$ and $f_{3}$ are single-peak functions, $f_{4}$ ~ $f_{10}$ are simple multi-peak functions, $f_{11}$ ~ $f_{20}$ are mixed functions, and $f_{21}$ ~ $f_{30}$ are compound functions. CEC17 functions are shown in Table 5.

The algorithms before and after the improvement are respectively referred to as PSO and PSOG. BBPSO and BBPSOG. The dimension of the experiment is 10, and the experiment was repeated 30 times. Parameter setting of the algorithm is shown in Table 6.

Considering that the swarm intelligence algorithm is a random algorithm, we repeated the experiments of each algorithm 30 times and recorded the optimal value, the worst value, the average value and the variance. Advantageous test function results are marked in bold. Statistical results are shown in Table 7, Table 8, Table 9 and Table 10.

In summary, the algorithm using a population diversity control strategy achieves a better optimization effect than the original algorithm in most test functions, which indicates that our strategy of “controlling the population diversity according to dimensional entropy to maintain diversity in the early stage while controlling convergence in the late stage is effective”. From the vertical perspective, the algorithm improved by the strategy has a more obvious optimization effect in the high dimension, which indicates that our concept of dimensional entropy is robust and adaptable in the high dimension.

## 5. Conclusions

This paper proposes a species diversity measure based on the dimension entropy mechanism, which creatively combines dimension learning and entropy. An independent view of the different dimensions of entropy affords strong robustness, while the entropy value method enjoys strong adaptability to changes in population size and an ability to have a clear response. Dimension entropy controls the population diversity of the swarm intelligence algorithm update strategy, which relies on the dimension entropy calculation of population diversity. By controlling the redundant control particle diversity, the slow algorithm, under the control of a lower diversity, improved the previous rapid loss of diversity in the swarm intelligence algorithm caused by local optimal results of stagnation of the algorithm, and maintained the diversity to ensure algorithm convergence. The 29 on the cec17 test function verified the effectiveness of this strategy.

## Figures and Tables

**Figure 1 entropy-23-00397-f001:**
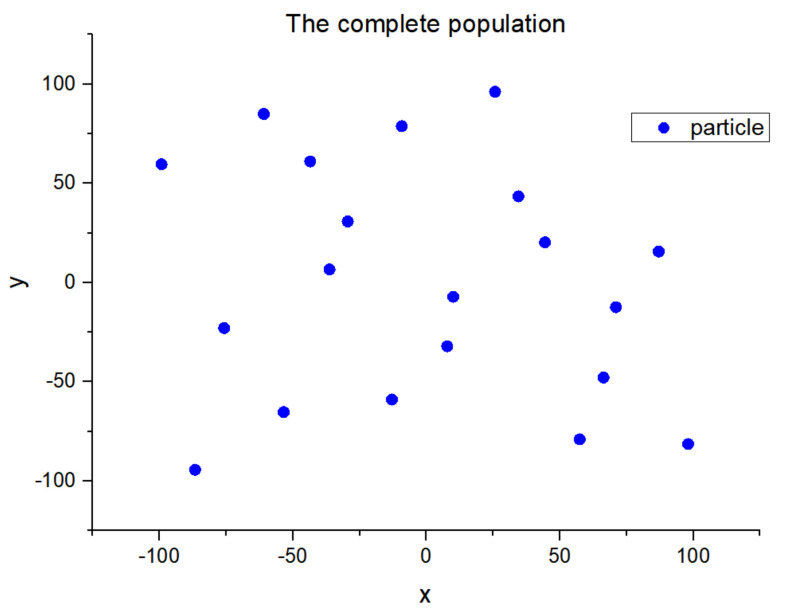
The complete population.

**Figure 2 entropy-23-00397-f002:**
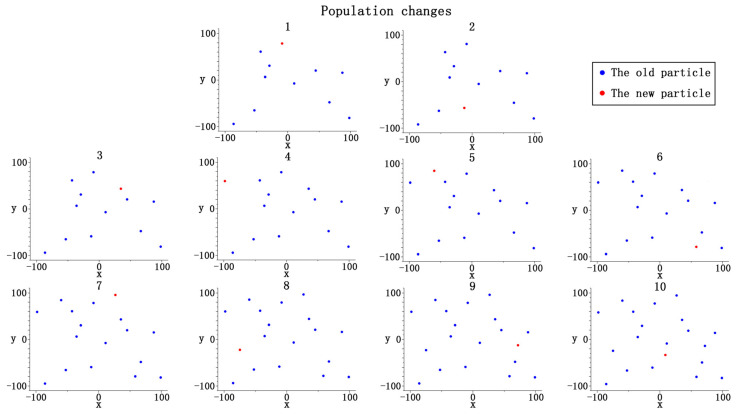
Schematic diagram of population change.

**Figure 3 entropy-23-00397-f003:**
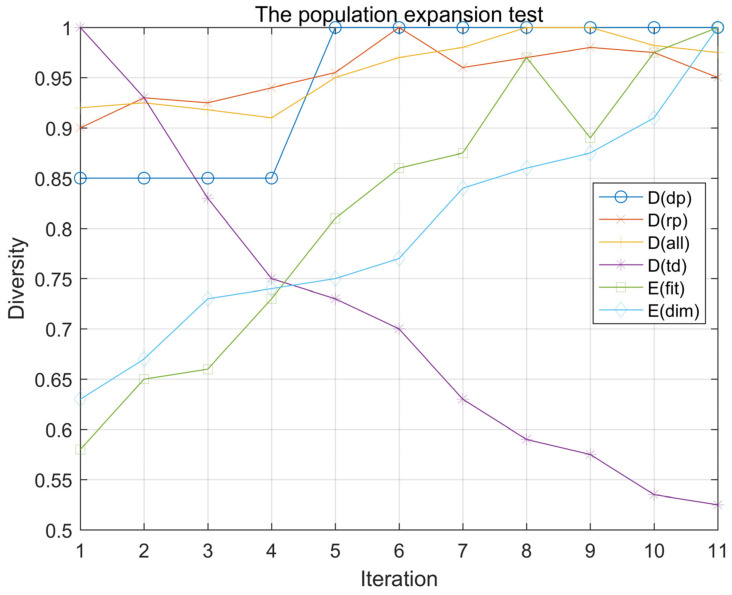
Results of population expansion experiments.

**Figure 4 entropy-23-00397-f004:**
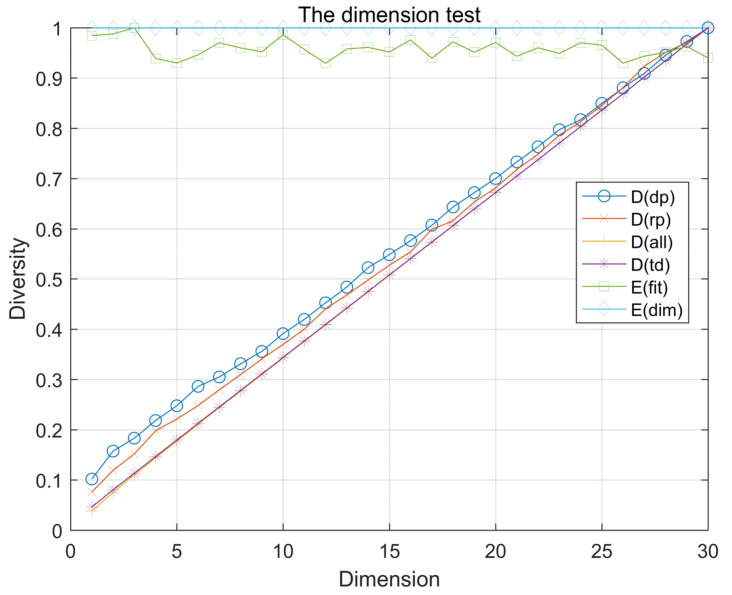
Dimensional robustness testing.

**Figure 5 entropy-23-00397-f005:**
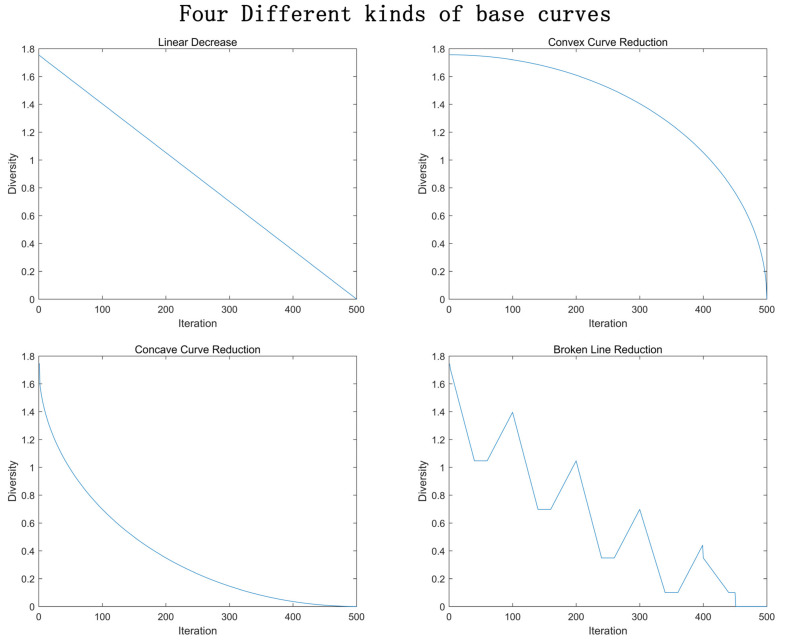
Four different kinds of base curves.

**Figure 6 entropy-23-00397-f006:**
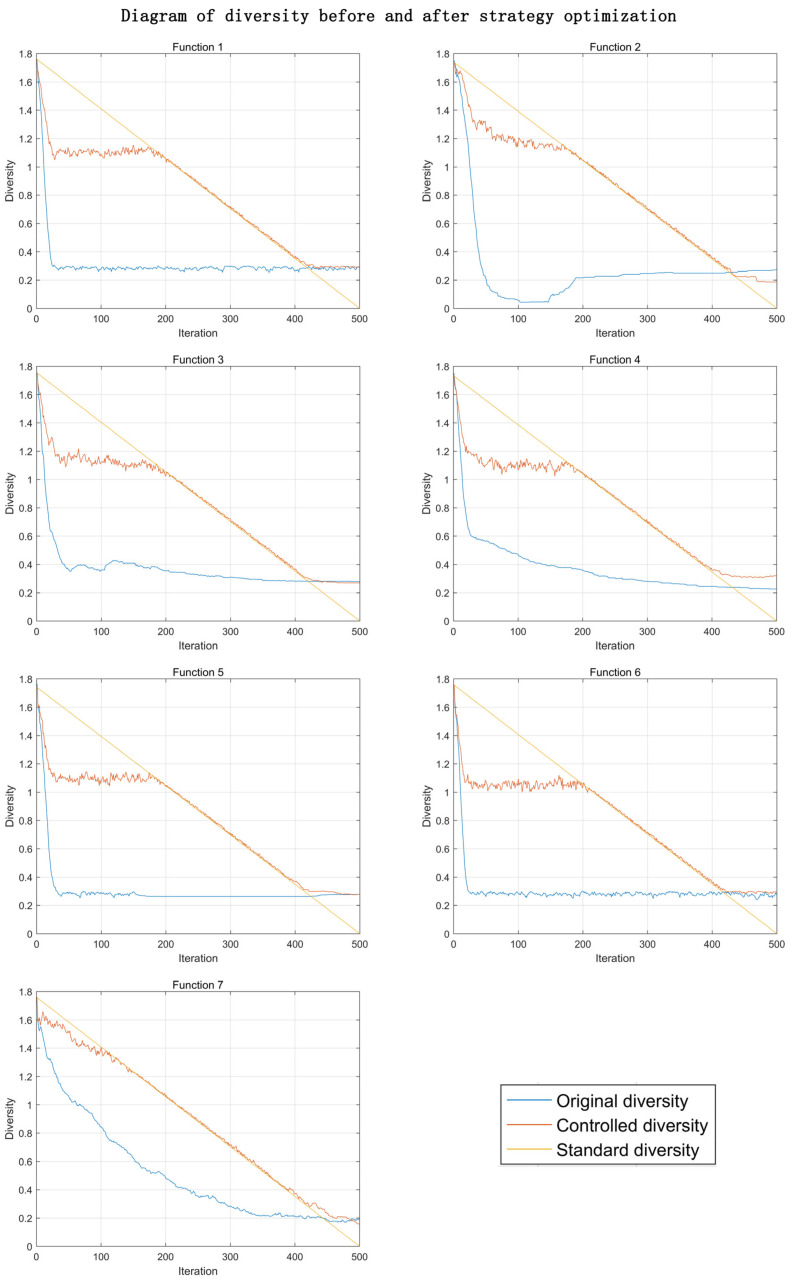
Comparison of diversity.

**Table 1 entropy-23-00397-t001:** Symbol summary.

Symbol	Definition
i,j	Variable
k	Dimension number ∈{1,2,…,n}
m	Interval number
M	Total number of intervals
n	Total number of dimension
$X_{i,k}$	The position of the i particle on the k dimension
$\overset{-}{X_{k}}$	Average value of the population on the k dimension
$p_{m,k}$	Fraction of N that belongs to interval m on the dimension k

**Table 2 entropy-23-00397-t002:** Rastrigin tests.

Time	$D_{dp}$	$D_{rp}$	$D_{all}$	$D_{td}$	$E_{fit}$	$E_{fuzzy}$	$E_{dim}$
1	0.782	0.692	0.997	0.983	0.967	0.779	0.977
2	0.925	0.855	1.000	0.980	0.950	0.889	0.984
3	0.708	0.889	0.997	0.973	0.947	0.840	0.976
4	0.715	0.642	0.997	0.978	0.942	0.758	0.977
5	0.821	0.714	0.998	0.986	0.961	0.747	0.989
6	0.931	0.852	0.992	0.942	0.951	0.875	0.979
7	0.949	0.618	1.000	0.990	0.978	0.907	0.980
8	0.925	0.844	0.998	0.989	0.975	0.907	0.980
9	0.799	0.699	0.996	0.976	0.952	0.805	0.984
10	0.818	0.712	0.937	0.982	0.953	0.807	0.972
Mean	0.837	0.752	0.991	0.978	0.958	0.831	0.980
Rank	5	7	1	3	4	6	2

**Table 3 entropy-23-00397-t003:** test functions. U:(unimodal),M:(multimodal).

Num	Function Name	Property	Best Value
1	Sphere’s Function	U	0
2	Rosenbrock’s Function	M	0
3	Rastrigin’s Function	M	0
4	Griewank’s Function	M	0
5	Ackley’s Function	M	0
6	Schwefel’s Problem 2.22	M	0
7	Schwefel’s Problem 1.2	M	0

**Table 4 entropy-23-00397-t004:** Comparison of the diversity and optimization results of the four curves.

No		Line	Convex Curve	Concave Curve	Broken Line
1	Min	**9.47 × 10^−57^**	3.26 × 10^−57^	6.86 × 10^−58^	6.69 × 10^−58^
	Max	**3.45 × 10^−51^**	6.43 × 10^−51^	2.48 × 10^2^	5.81 × 10^−48^
	Mean	**1.93 × 10^−52^**	3.51 × 10^−52^	8.27 × 10^0^	1.94 × 10^−49^
	DimEnt	**0.820**	1.039	0.574	0.756
2	Min	**8.83 × 10^−1^**	9.40 × 10^−1^	9.00 × 10^−1^	9.45 × 10^−1^
	Max	**4.80 × 10^1^**	2.26 × 10^2^	1.69 × 10^5^	6.99 × 10^2^
	Mean	**8.30 × 10^0^**	2.23 × 10^1^	5.65 × 10^3^	5.65 × 10^1^
	DimEnt	**0.831**	1.019	0.596	0.744
3	Min	**8.53 × 10^−14^**	9.95 × 10^−1^	0.00 × 10^0^	8.27 × 10^−12^
	Max	**6.81 × 10^0^**	6.96 × 10^0^	2.22 × 10^2^	4.97 × 10^0^
	Mean	**1.94 × 10^−0^**	2.89 × 10^0^	2.00 × 10^1^	2.04 × 10^0^
	DimEnt	**0.834**	1.039	0.626	0.771
4	Min	1.23 × 10^−2^	**1.23 × 10^−2^**	1.97 × 10^−2^	1.23 × 10^−2^
	Max	1.26 × 10^−1^	**1.43 × 10^−1^**	1.65 × 10^−1^	1.68 × 10^1^
	Mean	6.08 × 10^−2^	**5.83 × 10^−2^**	6.69 × 10^−2^	6.15 × 10^−1^
	DimEnt	0.792	**1.012**	0.540	0.750
5	Min	4.44 × 10^−15^	**4.44 × 10^−15^**	8.88 × 10^−16^	**4.44 × 10^−15^**
	Max	1.21 × 10^1^	**4.44 × 10^−15^**	1.42 × 10^1^	**4.44 × 10^−15^**
	Mean	4.03 × 10^−1^	**4.44 × 10^−15^**	8.08 × 10^−1^	**4.44 × 10^−15^**
	DimEnt	0.830	**1.036**	0.585	**0.753**
6	Min	3.98 × 10^−33^	**1.33 × 10^−32^**	1.09 × 10^−32^	3.11 × 10^−32^
	Max	1.20 × 10^−27^	**3.46 × 10^−29^**	1.32 × 10^1^	4.30 × 10^−29^
	Mean	4.34 × 10^−29^	**3.78 × 10^−30^**	1.71 × 10^0^	5.57 × 10^−30^
	DimEnt	0.818	**1.047**	0.664	0.751
7	Min	7.57 × 10^−1^	**6.30 × 10^−1^**	2.19 × 10^−2^	1.60 × 10^−1^
	Max	2.18 × 10^3^	**3.08 × 10^1^**	5.34 × 10^3^	1.23 × 10^3^
	Mean	9.04 × 10^1^	**7.58 × 10^0^**	5.60 × 10^2^	5.27 × 10^1^
	DimEnt	1.003	**1.188**	0.886	0.964

**Table 5 entropy-23-00397-t005:** CEC17 functions. U: (unimodal),M: (multimodal),H: (hybrid),C: (composition).

Num	Function Name	Property	Best Value
F01	Shifted and Rotated Bent Cigar Function	U	100
F03	Shifted and Rotated Zakharov Function	M	300
F04	Shifted and Rotated Rosenbrock’s Function	M	400
F05	Shifted and Rotated Rastrigin’s Function	M	500
F06	Shifted and Rotated Expanded Scaffer’s F6 Function	M	600
F07	Shifted and Rotated Lunacek Bi_Rastrigin Function	M	700
F08	Shifted and Rotated Non-Continuous Rastrigin’s Function	M	800
F09	Shifted and Rotated Levy Function	M	900
F10	Shifted and Rotated Schwefel’s Function	M	1000
F11	Hybrid Function 1 (N = 3)	H	1100
F12	Hybrid Function 2 (N = 3)	H	1200
F13	Hybrid Function 3 (N = 3)	H	1300
F14	Hybrid Function 4 (N = 4)	H	1400
F15	Hybrid Function 5 (N = 4)	H	1500
F16	Hybrid Function 6 (N = 4)	H	1600
F17	Hybrid Function 6 (N = 5)	H	1700
F18	Hybrid Function 6 (N = 5)	H	1800
F19	Hybrid Function 6 (N = 5)	H	1900
F20	Hybrid Function 6 (N = 6)	H	2000
F21	Composition Function 1	C	2100
F22	Composition Function 2	C	2200
F23	Composition Function 3	C	2300
F24	Composition Function 4	C	2400
F25	Composition Function 5	C	2500
F26	Composition Function 6	C	2600
F27	Composition Function 7	C	2700
F28	Composition Function 8	C	2800
F29	Composition Function 9	C	2900
F30	Composition Function 10	C	3000

**Table 6 entropy-23-00397-t006:** Parameter setting.

Algorithm	Parameter Setting
PSO	ω:0.5, $c_{1}=c_{2}=2$

**Table 7 entropy-23-00397-t007:** PSO 10-dimensional improvement comparison.

fun	PSO(Dim = 10)				PSOG(Dim = 10)			
	min	max	mean	std	min	max	mean	std
$f_{1}$	1.02 × 10^2^	2.54 × 10^3^	1.16 × 10^3^	8.87 × 10^2^	**1.01 × 10^2^**	**1.53 × 10^3^**	**5.89 × 10^2^**	**4.20 × 10^2^**
$f_{3}$	3.00 × 10^2^	3.00 × 10^2^	3.00 × 10^2^	0.00 × 10^0^	3.00 × 10^2^	3.00 × 10^2^	3.00 × 10^2^	0.00 × 10^0^
$f_{4}$	4.00 × 10^2^	4.35 × 10^2^	4.26 × 10^2^	1.46 × 10^1^	**4.00 × 10^2^**	**4.35 × 10^2^**	**4.09 × 10^2^**	**1.36 × 10^1^**
$f_{5}$	5.07 × 10^2^	5.34 × 10^2^	5.18 × 10^2^	6.35 × 10^0^	**5.04 × 10^2^**	**5.17 × 10^2^**	**5.13 × 10^2^**	**3.95 × 10^0^**
$f_{6}$	6.00 × 10^2^	6.07 × 10^2^	6.00 × 10^2^	1.22 × 10^0^	6.00 × 10^2^	6.00 × 10^2^	6.00 × 10^2^	0.00 × 10^0^
$f_{7}$	7.13 × 10^2^	7.38 × 10^2^	7.21 × 10^2^	5.54 × 10^0^	**7.12 × 10^2^**	**7.21 × 10^2^**	**7.18 × 10^2^**	**2.12 × 10^0^**
$f_{8}$	8.06 × 10^2^	8.36 × 10^2^	8.16 × 10^2^	6.94 × 10^0^	**8.07 × 10^2^**	**8.21 × 10^2^**	**8.14 × 10^2^**	**4.62 × 10^0^**
$f_{9}$	9.00 × 10^2^	9.00 × 10^2^	9.00 × 10^2^	1.63 × 10^−2^	9.00 × 10^2^	9.00 × 10^2^	9.00 × 10^2^	0.00 × 10^0^
$f_{10}$	1.13 × 10^3^	1.85 × 10^3^	1.48 × 10^3^	1.96 × 10^2^	**1.13 × 10^3^**	**1.45 × 10^3^**	**1.31 × 10^3^**	**9.45 × 10^1^**
$f_{11}$	1.10 × 10^3^	1.14 × 10^3^	1.12 × 10^3^	8.54 × 10^0^	**1.10 × 10^3^**	**1.12 × 10^3^**	**1.11 × 10^3^**	**4.58 × 10^0^**
$f_{12}$	2.05 × 10^3^	4.35 × 10^5^	2.75 × 10^4^	7.79 × 10^4^	**1.88 × 10^3^**	**2.13 × 10^4^**	**9.67 × 10^3^**	**6.94 × 10^3^**
$f_{13}$	1.34 × 10^3^	9.10 × 10^3^	3.39 × 10^3^	2.16 × 10^3^	**1.31 × 10^3^**	**3.43 × 10^3^**	**2.06 × 10^3^**	**6.48 × 10^2^**
$f_{14}$	1.43 × 10^3^	1.77 × 10^3^	1.49 × 10^3^	6.32 × 10^1^	**1.44 × 10^3^**	**1.47 × 10^3^**	**1.45 × 10^3^**	**7.40 × 10^0^**
$f_{15}$	1.51 × 10^3^	1.76 × 10^3^	1.56 × 10^3^	6.06 × 10^1^	**1.51 × 10^3^**	**1.53 × 10^3^**	**1.52 × 10^3^**	**5.95 × 10^0^**
$f_{16}$	1.60 × 10^3^	1.86 × 10^3^	1.72 × 10^3^	6.14 × 10^1^	**1.60 × 10^3^**	**1.72 × 10^3^**	**1.64 × 10^3^**	**5.27 × 10^1^**
$f_{17}$	1.73 × 10^3^	1.78 × 10^3^	1.75 × 10^3^	1.39 × 10^1^	**1.71 × 10^3^**	**1.75 × 10^3^**	**1.73 × 10^3^**	**8.66 × 10^0^**
$f_{18}$	1.84 × 10^3^	1.29 × 10^4^	5.07 × 10^3^	2.91 × 10^3^	**1.93 × 10^3^**	**5.67 × 10^3^**	**3.66 × 10^3^**	**1.04 × 10^3^**
$f_{19}$	1.90 × 10^3^	1.96 × 10^3^	1.92 × 10^3^	1.07 × 10^1^	**1.91 × 10^3^**	**1.92 × 10^3^**	**1.91 × 10^3^**	**3.93 × 10^0^**
$f_{20}$	2.01 × 10^3^	2.20 × 10^3^	2.07 × 10^3^	5.31 × 10^1^	**2.00 × 10^3^**	**2.04 × 10^3^**	**2.03 × 10^3^**	**7.21 × 10^0^**
$f_{21}$	2.20 × 10^3^	2.20 × 10^3^	2.20 × 10^3^	2.67 × 10^−13^	2.20 × 10^3^	2.20 × 10^3^	2.20 × 10^3^	2.09 × 10^−13^
$f_{22}$	2.30 × 10^3^	2.30 × 10^3^	2.30 × 10^3^	2.39 × 10^−13^	2.21 × 10^3^	2.30 × 10^3^	2.30 × 10^3^	2.06 × 10^1^
$f_{23}$	2.40 × 10^3^	2.82 × 10^3^	2.71 × 10^3^	7.80 × 10^1^	**2.40 × 10^3^**	**2.67 × 10^3^**	**2.62 × 10^3^**	**9.54 × 10^1^**
$f_{24}$	2.50 × 10^3^	2.80 × 10^3^	2.61 × 10^3^	5.79 × 10^1^	**2.50 × 10^3^**	**2.60 × 10^3^**	**2.59 × 10^3^**	**3.08 × 10^1^**
$f_{25}$	2.89 × 10^3^	2.95 × 10^3^	2.94 × 10^3^	2.04 × 10^1^	**2.90 × 10^3^**	**2.95 × 10^3^**	**2.93 × 10^3^**	**2.32 × 10^1^**
$f_{26}$	2.80 × 10^3^	3.49 × 10^3^	2.94 × 10^3^	2.04 × 10^2^	**2.60 × 10^3^**	**2.90 × 10^3^**	**2.83 × 10^3^**	**7.33 × 10^1^**
$f_{27}$	3.10 × 10^3^	3.50 × 10^3^	3.29 × 10^3^	1.15 × 10^2^	**3.10 × 10^3^**	**3.23 × 10^3^**	**3.16 × 10^3^**	**4.45 × 10^1^**
$f_{28}$	3.10 × 10^3^	3.23 × 10^3^	3.15 × 10^3^	2.52 × 10^1^	**3.10 × 10^3^**	**3.15 × 10^3^**	**3.13 × 10^3^**	**2.40 × 10^1^**
$f_{29}$	3.15 × 10^3^	3.30 × 10^3^	3.18 × 10^3^	3.45 × 10^1^	**3.14 × 10^3^**	**3.17 × 10^3^**	**3.16 × 10^3^**	**1.05 × 10^1^**
$f_{30}$	3.49 × 10^3^	3.85 × 10^4^	9.59 × 10^3^	7.45 × 10^3^	**3.71 × 10^3^**	**7.49 × 10^3^**	**5.22 × 10^3^**	**1.13 × 10^3^**
count	0				24			

**Table 8 entropy-23-00397-t008:** PSO 30-dimensional improvement comparison.

fun	PSO(Dim = 30)				PSOG(Dim = 30)			
	min	max	mean	std	min	max	mean	std
$f_{1}$	1.00 × 10^2^	1.22 × 10^9^	1.37 × 10^8^	3.26 × 10^8^	**1.00 × 10^2^**	**1.01 × 10^2^**	**1.00 × 10^2^**	**2.95 × 10^−1^**
$f_{3}$	3.05 × 10^2^	3.93 × 10^2^	3.34 × 10^2^	2.29 × 10^1^	**3.09 × 10^2^**	**3.53 × 10^2^**	**3.31 × 10^2^**	**1.51 × 10^1^**
$f_{4}$	4.00 × 10^2^	6.38 × 10^2^	4.89 × 10^2^	5.06 × 10^1^	**4.04 × 10^2^**	**4.71 × 10^2^**	**4.66 × 10^2^**	**1.48 × 10^1^**
$f_{5}$	5.73 × 10^2^	6.71 × 10^2^	6.05 × 10^2^	2.42 × 10^1^	**5.64 × 10^2^**	**5.95 × 10^2^**	**5.80 × 10^2^**	**9.27 × 10^0^**
$f_{6}$	6.00 × 10^2^	6.23 × 10^2^	6.08 × 10^2^	5.93 × 10^0^	**6.00 × 10^2^**	**6.08 × 10^2^**	**6.04 × 10^2^**	**2.85 × 10^0^**
$f_{7}$	7.68 × 10^2^	8.46 × 10^2^	8.10 × 10^2^	2.12 × 10^1^	**7.77 × 10^2^**	**8.22 × 10^2^**	**7.99 × 10^2^**	**1.40 × 10^1^**
$f_{8}$	8.67 × 10^2^	9.89 × 10^2^	9.18 × 10^2^	2.93 × 10^1^	**8.75 × 10^2^**	**9.39 × 10^2^**	**9.09 × 10^2^**	**1.90 × 10^1^**
$f_{9}$	9.08 × 10^2^	4.85 × 10^3^	2.61 × 10^3^	1.02 × 10^3^	**9.31 × 10^2^**	**2.88 × 10^3^**	**1.87 × 10^3^**	**6.15 × 10^2^**
$f_{10}$	2.80 × 10^3^	5.20 × 10^3^	4.07 × 10^3^	6.36 × 10^2^	**2.96 × 10^3^**	**4.15 × 10^3^**	**3.63 × 10^3^**	**3.18 × 10^2^**
$f_{11}$	1.18 × 10^3^	1.43 × 10^3^	1.25 × 10^3^	5.91 × 10^1^	**1.17 × 10^3^**	**1.27 × 10^3^**	**1.23 × 10^3^**	**3.01 × 10^1^**
$f_{12}$	2.64 × 10^3^	3.35 × 10^8^	1.12 × 10^7^	6.12 × 10^7^	**3.63 × 10^3^**	**1.50 × 10^4^**	**7.33 × 10^3^**	**3.51 × 10^2^**
$f_{13}$	1.35 × 10^3^	1.02 × 10^4^	2.05 × 10^3^	1.80 × 10^3^	**1.41 × 10^3^**	**2.71 × 10^3^**	**1.85 × 10^3^**	**4.06 × 10^2^**
$f_{14}$	1.48 × 10^3^	1.97 × 10^3^	1.68 × 10^3^	1.12 × 10^2^	**1.53 × 10^3^**	**1.71 × 10^3^**	**1.64 × 10^3^**	**5.51 × 10^1^**
$f_{15}$	1.53 × 10^3^	1.92 × 10^3^	1.59 × 10^3^	7.02 × 10^1^	**1.52 × 10^3^**	**1.61 × 10^3^**	**1.57 × 10^3^**	**2.55 × 10^1^**
$f_{16}$	1.86 × 10^3^	2.91 × 10^3^	2.35 × 10^3^	2.68 × 10^2^	**1.96 × 10^3^**	**2.44 × 10^3^**	**2.23 × 10^3^**	**1.67 × 10^2^**
$f_{17}$	1.80 × 10^3^	2.52 × 10^3^	2.09 × 10^3^	1.83 × 10^2^	**1.79 × 10^3^**	**2.02 × 10^3^**	**1.89 × 10^3^**	**7.44 × 10^1^**
$f_{18}$	5.96 × 10^3^	1.26 × 10^5^	3.88 × 10^4^	2.60 × 10^4^	**9.55 × 10^3^**	**4.17 × 10^4^**	**2.33 × 10^4^**	**1.11 × 10^4^**
$f_{19}$	1.98 × 10^3^	2.93 × 10^4^	6.50 × 10^3^	6.01 × 10^3^	**1.95 × 10^3^**	**4.41 × 10^3^**	**2.77 × 10^3^**	**8.62 × 10^2^**
$f_{20}$	2.20 × 10^3^	2.71 × 10^3^	2.42 × 10^3^	1.08 × 10^2^	**2.13 × 10^3^**	**2.44 × 10^3^**	**2.31 × 10^3^**	**1.05 × 10^2^**
$f_{21}$	**2.20 × 10^3^**	**2.20 × 10^3^**	**2.20 × 10^3^**	**2.25 × 10^−1^**	2.25 × 10^3^	2.25 × 10^3^	2.25 × 10^3^	4.67 × 10^−13^
$f_{22}$	**2.30 × 10^3^**	**2.30 × 10^3^**	**2.30 × 10^3^**	**2.24 × 10^−1^**	2.35 × 10^3^	2.35 × 10^3^	2.35 × 10^3^	4.55 × 10^−13^
$f_{23}$	3.04 × 10^3^	4.20 × 10^3^	3.54 × 10^3^	3.24 × 10^2^	**2.83 × 10^3^**	**2.88 × 10^3^**	**2.87 × 10^3^**	**1.30 × 10^1^**
$f_{24}$	2.60 × 10^3^	2.61 × 10^3^	2.60 × 10^3^	1.55 × 10^0^	2.60 × 10^3^	2.60 × 10^3^	2.60 × 10^3^	4.43 × 10^−13^
$f_{25}$	2.90 × 10^3^	3.05 × 10^3^	2.94 × 10^3^	4.21 × 10^1^	2.90 × 10^3^	2.97 × 10^3^	2.94 × 10^3^	2.79 × 10^1^
$f_{26}$	2.80 × 10^3^	2.90 × 10^3^	2.80 × 10^3^	1.83 × 10^1^	2.80 × 10^3^	2.80 × 10^3^	2.80 × 10^3^	5.00 × 10^−13^
$f_{27}$	3.78 × 10^3^	5.06 × 10^3^	4.39 × 10^3^	3.41 × 10^2^	**3.38 × 10^3^**	**3.59 × 10^3^**	**3.51 × 10^3^**	**5.25 × 10^1^**
$f_{28}$	3.17 × 10^3^	3.95 × 10^3^	3.31 × 10^3^	1.39 × 10^2^	**3.17 × 10^3^**	**3.28 × 10^3^**	**3.24 × 10^3^**	**3.29 × 10^1^**
$f_{29}$	3.35 × 10^3^	4.11 × 10^3^	3.59 × 10^3^	2.12 × 10^2^	**3.29 × 10^3^**	**3.65 × 10^3^**	**3.49 × 10^3^**	**1.10 × 10^2^**
$f_{30}$	4.19 × 10^3^	1.88 × 10^5^	1.60 × 10^4^	3.39 × 10^4^	**4.44 × 10^3^**	**1.60 × 10^4^**	**9.79 × 10^3^**	**3.31 × 10^3^**
count	2				24			

**Table 9 entropy-23-00397-t009:** BBPSO 10-dimensional improvement comparison.

fun	BBPSO(Dim = 10)				BBPSOG(Dim = 10)			
	min	max	mean	std	min	max	mean	std
$f_{1}$	1.28 × 10^2^	2.54 × 10^3^	1.28 × 10^3^	6.85 × 10^2^	**1.50 × 10^2^**	**2.12 × 10^3^**	**1.21 × 10^3^**	**5.45 × 10^2^**
$f_{3}$	3.00 × 10^2^	3.00 × 10^2^	3.00 × 10^2^	0.00 × 10^0^	3.00 × 10^2^	3.00 × 10^2^	3.00 × 10^2^	0.00 × 10^0^
$f_{4}$	4.00 × 10^2^	5.21 × 10^2^	4.30 × 10^2^	2.23 × 10^1^	**4.00 × 10^2^**	**4.35 × 10^2^**	**4.17 × 10^2^**	**1.65 × 10^1^**
$f_{5}$	5.04 × 10^2^	5.27 × 10^2^	5.13 × 10^2^	5.83 × 10^0^	**5.05 × 10^2^**	**5.12 × 10^2^**	**5.09 × 10^2^**	**2.08 × 10^0^**
$f_{6}$	6.00 × 10^2^	6.01 × 10^2^	6.00 × 10^2^	1.76 × 10^−1^	6.00 × 10^2^	6.00 × 10^2^	6.00 × 10^2^	3.69 × 10^−14^
$f_{7}$	7.08 × 10^2^	7.26 × 10^2^	7.18 × 10^2^	4.38 × 10^0^	7.13 × 10^2^	7.22 × 10^2^	7.18 × 10^2^	2.75 × 10^0^
$f_{8}$	8.05 × 10^2^	8.22 × 10^2^	8.12 × 10^2^	4.40 × 10^0^	**8.05 × 10^2^**	**8.13 × 10^2^**	**8.09 × 10^2^**	**2.63 × 10^0^**
$f_{9}$	9.00 × 10^2^	9.02 × 10^2^	9.00 × 10^2^	4.54 × 10^−1^	9.00 × 10^2^	9.00 × 10^2^	9.00 × 10^2^	0.00 × 10^0^
$f_{10}$	1.03 × 10^3^	1.77 × 10^3^	1.34 × 10^3^	2.04 × 10^2^	**1.04 × 10^3^**	**1.35 × 10^3^**	**1.17 × 10^3^**	**1.10 × 10^2^**
$f_{11}$	1.10 × 10^3^	1.12 × 10^3^	1.11 × 10^3^	5.17 × 10^0^	**1.10 × 10^3^**	**1.11 × 10^3^**	**1.10 × 10^3^**	**1.91 × 10^0^**
$f_{12}$	2.40 × 10^3^	4.36 × 10^5^	3.59 × 10^4^	7.77 × 10^4^	**3.97 × 10^3^**	**2.58 × 10^4^**	**1.19 × 10^4^**	**5.72 × 10^3^**
$f_{13}$	1.31 × 10^3^	9.37 × 10^3^	4.41 × 10^3^	2.86 × 10^3^	**1.32 × 10^3^**	**4.04 × 10^3^**	**2.27 × 10^3^**	**9.62 × 10^2^**
$f_{14}$	1.43 × 10^3^	1.54 × 10^3^	1.46 × 10^3^	2.87 × 10^1^	**1.43 × 10^3^**	**1.44 × 10^3^**	**1.43 × 10^3^**	**5.87 × 10^0^**
$f_{15}$	1.51 × 10^3^	1.69 × 10^3^	1.59 × 10^3^	5.23 × 10^1^	**1.51 × 10^3^**	**1.59 × 10^3^**	**1.54 × 10^3^**	**2.34 × 10^1^**
$f_{16}$	1.60 × 10^3^	1.81 × 10^3^	1.68 × 10^3^	7.19 × 10^1^	**1.60 × 10^3^**	**1.64 × 10^3^**	**1.61 × 10^3^**	**1.26 × 10^1^**
$f_{17}$	1.71 × 10^3^	1.85 × 10^3^	1.75 × 10^3^	3.17 × 10^1^	**1.72 × 10^3^**	**1.74 × 10^3^**	**1.73 × 10^3^**	**5.67 × 10^0^**
$f_{18}$	1.86 × 10^3^	2.34 × 10^4^	6.48 × 10^3^	5.58 × 10^3^	**2.09 × 10^3^**	**5.37 × 10^3^**	**3.03 × 10^3^**	**8.68 × 10^2^**
$f_{19}$	1.90 × 10^3^	2.12 × 10^3^	1.94 × 10^3^	5.16 × 10^1^	**1.90 × 10^3^**	**1.92 × 10^3^**	**1.91 × 10^3^**	**4.37 × 10^0^**
$f_{20}$	2.00 × 10^3^	2.08 × 10^3^	2.03 × 10^3^	1.79 × 10^1^	2.00 × 10^3^	2.04 × 10^3^	2.03 × 10^3^	1.05 × 10^1^
$f_{21}$	**2.20 × 10^3^**	**2.20 × 10^3^**	**2.20 × 10^3^**	**2.53 × 10^−13^**	2.25 × 10^3^	2.27 × 10^3^	2.26 × 10^3^	7.21 × 10^0^
$f_{22}$	**2.21 × 10^3^**	**2.30 × 10^3^**	**2.30 × 10^3^**	**1.70 × 10^1^**	2.24 × 10^3^	2.39 × 10^3^	2.35 × 10^3^	4.83 × 10^1^
$f_{23}$	2.65 × 10^3^	2.71 × 10^3^	2.68 × 10^3^	1.31 × 10^1^	**2.65 × 10^3^**	**2.67 × 10^3^**	**2.67 × 10^3^**	**5.07 × 10^0^**
$f_{24}$	2.50 × 10^3^	2.82 × 10^3^	2.74 × 10^3^	1.21 × 10^2^	**2.50 × 10^3^**	**2.81 × 10^3^**	**2.72 × 10^3^**	**1.39 × 10^2^**
$f_{25}$	2.89 × 10^3^	2.97 × 10^3^	2.93 × 10^3^	2.54 × 10^1^	**2.89 × 10^3^**	**2.94 × 10^3^**	**2.91 × 10^3^**	**1.86 × 10^1^**
$f_{26}$	2.90 × 10^3^	3.62 × 10^3^	3.21 × 10^3^	2.42 × 10^2^	**2.60 × 10^3^**	**3.37 × 10^3^**	**3.00 × 10^3^**	**1.92 × 10^2^**
$f_{27}$	3.14 × 10^3^	3.31 × 10^3^	3.17 × 10^3^	4.30 × 10^1^	**3.12 × 10^3^**	**3.15 × 10^3^**	**3.14 × 10^3^**	**5.87 × 10^0^**
$f_{28}$	3.10 × 10^3^	3.37 × 10^3^	3.18 × 10^3^	6.51 × 10^1^	**3.10 × 10^3^**	**3.15 × 10^3^**	**3.13 × 10^3^**	**2.50 × 10^1^**
$f_{29}$	3.14 × 10^3^	3.29 × 10^3^	3.18 × 10^3^	3.12 × 10^1^	**3.14 × 10^3^**	**3.17 × 10^3^**	**3.16 × 10^3^**	**7.45 × 10^0^**
$f_{30}$	3.97 × 10^3^	2.32 × 10^5^	1.58 × 10^4^	4.10 × 10^4^	**4.66 × 10^3^**	**1.03 × 10^4^**	**7.68 × 10^3^**	**1.52 × 10^3^**
count	2				22			

**Table 10 entropy-23-00397-t010:** BBPSO 30-dimensional improvement comparison.

fun	BBPSO(Dim = 30)				BBPSOG(Dim = 30)			
	min	max	mean	std	min	max	mean	std
$f_{1}$	1.00 × 10^2^	5.10 × 10^9^	1.58 × 10^9^	1.52 × 10^9^	**1.00 × 10^2^**	**2.91 × 10^3^**	**1.22 × 10^3^**	**1.41 × 10^3^**
$f_{3}$	9.72 × 10^3^	3.63 × 10^4^	2.08 × 10^4^	6.19 × 10^3^	**6.85 × 10^3^**	**2.74 × 10^4^**	**2.05 × 10^4^**	**5.65 × 10^3^**
$f_{4}$	4.06 × 10^2^	9.89 × 10^2^	6.01 × 10^2^	1.48 × 10^2^	**4.63 × 10^2^**	**4.76 × 10^2^**	**4.69 × 10^2^**	**4.71 × 10^0^**
$f_{5}$	5.52 × 10^2^	7.29 × 10^2^	6.31 × 10^2^	3.50 × 10^1^	**5.51 × 10^2^**	**5.91 × 10^2^**	**5.76 × 10^2^**	**1.04 × 10^1^**
$f_{6}$	6.00 × 10^2^	6.26 × 10^2^	6.06 × 10^2^	5.35 × 10^0^	**6.00 × 10^2^**	**6.01 × 10^2^**	**6.00 × 10^2^**	**2.04 × 10^−1^**
$f_{7}$	7.72 × 10^2^	9.96 × 10^2^	8.53 × 10^2^	5.20 × 10^1^	**7.81 × 10^2^**	**8.34 × 10^2^**	**8.16 × 10^2^**	**1.54 × 10^1^**
$f_{8}$	8.69 × 10^2^	1.02 × 10^3^	9.28 × 10^2^	3.52 × 10^1^	**8.53 × 10^2^**	**8.85 × 10^2^**	**8.71 × 10^2^**	**1.00 × 10^1^**
$f_{9}$	1.39 × 10^3^	1.06 × 10^4^	2.93 × 10^3^	1.90 × 10^3^	**9.16 × 10^2^**	**1.22 × 10^3^**	**1.03 × 10^3^**	**9.76 × 10^1^**
$f_{10}$	2.63 × 10^3^	5.80 × 10^3^	4.24 × 10^3^	7.55 × 10^2^	**2.76 × 10^3^**	**3.82 × 10^3^**	**3.40 × 10^3^**	**2.87 × 10^2^**
$f_{11}$	1.21 × 10^3^	1.97 × 10^3^	1.43 × 10^3^	1.49 × 10^2^	**1.18 × 10^3^**	**1.30 × 10^3^**	**1.25 × 10^3^**	**3.68 × 10^1^**
$f_{12}$	2.25 × 10^4^	5.70 × 10^8^	5.68 × 10^7^	1.31 × 10^8^	**6.28 × 10^3^**	**9.46 × 10^4^**	**3.70 × 10^4^**	**2.61 × 10^4^**
$f_{13}$	1.92 × 10^3^	9.56 × 10^5^	4.38 × 10^4^	1.73 × 10^5^	**3.58 × 10^3^**	**1.03 × 10^4^**	**6.82 × 10^3^**	**1.86 × 10^3^**
$f_{14}$	1.46 × 10^3^	1.38 × 10^6^	4.90 × 10^4^	2.51 × 10^5^	**1.50 × 10^3^**	**1.64 × 10^3^**	**1.55 × 10^3^**	**4.16 × 10^1^**
$f_{15}$	1.88 × 10^3^	3.36 × 10^4^	9.47 × 10^3^	8.20 × 10^3^	**1.83 × 10^3^**	**3.81 × 10^3^**	**2.37 × 10^3^**	**4.74 × 10^2^**
$f_{16}$	1.83 × 10^3^	3.09 × 10^3^	2.53 × 10^3^	3.53 × 10^2^	**1.97 × 10^3^**	**2.53 × 10^3^**	**2.28 × 10^3^**	**1.73 × 10^2^**
$f_{17}$	1.89 × 10^3^	2.51 × 10^3^	2.07 × 10^3^	1.41 × 10^2^	**1.87 × 10^3^**	**2.06 × 10^3^**	**1.95 × 10^3^**	**5.21 × 10^1^**
$f_{18}$	2.58 × 10^4^	7.94 × 10^5^	1.65 × 10^5^	1.55 × 10^5^	**2.09 × 10^4^**	**1.25 × 10^5^**	**7.43 × 10^4^**	**2.93 × 10^4^**
$f_{19}$	2.02 × 10^3^	4.68 × 10^4^	1.18 × 10^4^	1.19 × 10^4^	**2.00 × 10^3^**	**1.23 × 10^4^**	**5.57 × 10^3^**	**3.89 × 10^3^**
$f_{20}$	2.09 × 10^3^	2.57 × 10^3^	2.30 × 10^3^	1.22 × 10^2^	**2.09 × 10^3^**	**2.29 × 10^3^**	**2.21 × 10^3^**	**6.24 × 10^1^**
$f_{21}$	2.10 × 10^3^	2.82 × 10^3^	2.25 × 10^3^	1.33 × 10^2^	2.25 × 10^3^	2.25 × 10^3^	2.25 × 10^3^	4.30 × 10^−13^
$f_{22}$	**2.26 × 10^3^**	**2.40 × 10^3^**	**2.31 × 10^3^**	**3.30 × 10^1^**	2.35 × 10^3^	2.35 × 10^3^	2.35 × 10^3^	4.55 × 10^−13^
$f_{23}$	2.88 × 10^3^	3.06 × 10^3^	2.93 × 10^3^	3.57 × 10^1^	**2.85 × 10^3^**	**2.88 × 10^3^**	**2.86 × 10^3^**	**8.17 × 10^0^**
$f_{24}$	3.43 × 10^3^	3.55 × 10^3^	3.47 × 10^3^	3.19 × 10^1^	**3.38 × 10^3^**	**3.41 × 10^3^**	**3.40 × 10^3^**	**9.35 × 10^0^**
$f_{25}$	2.90 × 10^3^	3.25 × 10^3^	3.00 × 10^3^	8.65 × 10^1^	**2.91 × 10^3^**	**2.98 × 10^3^**	**2.93 × 10^3^**	**2.68 × 10^1^**
$f_{26}$	5.26 × 10^3^	6.77 × 10^3^	5.93 × 10^3^	3.56 × 10^2^	**3.54 × 10^3^**	**5.50 × 10^3^**	**5.13 × 10^3^**	**5.39 × 10^2^**
$f_{27}$	3.44 × 10^3^	3.79 × 10^3^	3.58 × 10^3^	8.82 × 10^1^	**3.41 × 10^3^**	**3.50 × 10^3^**	**3.46 × 10^3^**	**2.89 × 10^1^**
$f_{28}$	3.28 × 10^3^	5.56 × 10^3^	4.34 × 10^3^	9.07 × 10^2^	**3.18 × 10^3^**	**5.16 × 10^3^**	**3.80 × 10^3^**	**9.12 × 10^2^**
$f_{29}$	3.41 × 10^3^	4.13 × 10^3^	3.70 × 10^3^	1.82 × 10^2^	**3.31 × 10^3^**	**3.57 × 10^3^**	**3.46 × 10^3^**	**7.66 × 10^1^**
$f_{30}$	7.84 × 10^3^	9.17 × 10^5^	1.52 × 10^5^	2.42 × 10^5^	**5.25 × 10^3^**	**4.65 × 10^4^**	**2.07 × 10^4^**	**1.24 × 10^4^**
count	1				27			

## Data Availability

Not applicable.

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
