# Peer review of "A Diversity Model Based on Dimension Entropy and Its Application to Swarm Intelligence Algorithm"

_entropy, 2021, doi:10.3390/e23040397_

Round 1

Reviewer 2 Report

General comment:

The article is poorly written in some parts. Many paragraphs or sentences need to be rewritten for better readability. The control mechanism based on the dimension entropy proposed by the authors is simple: a comparison should be included when the authors’ mechanism is included in a particular evolutionary algorithm (EA) and compared to other variants of the same EA with self-control of its parameters.

Several remarks:

• Introduction, second paragraph: Although it is difficult to differentiate or classify an algorithm between swarm intelligence and evolutionary algorithm, I would not classify classic genetic algorithms (GA) and differential evolution (DE) as "swarm-based" (although some authors do). In these EAs (GAs and DE) there is no information exchange or cooperation between the coded solutions to obtain a collective behavior, beyond the comparison of fitness for the selection process and after their corresponding genetic operators, as in any EA. On the contrary, there are more typical examples of swarm-intelligence based-algorithms, usually inspired by the collective behavior of an organized group of animals and not mentioned by the authors: bat-algorithms, bee algorithms, firefly algorithm, bacterial foraging optimization, ….

Some previous studies that introduce the use of entropy in some aspect of the evolutionary/swarm algorithms are not included in the commented references. With a simple search, several articles are found, such as:

Folino G., Forestiero A. (2010), "Using Entropy for Evaluating Swarm Intelligence Algorithms". In: González J.R., Pelta D.A., Cruz C., Terrazas G., Krasnogor N. (eds) Nature Inspired Cooperative Strategies for Optimization (NICSO 2010). Studies in Computational Intelligence, vol 284. Springer, Berlin, Heidelberg. https://doi.org/10.1007/978-3-642-12538-6_28

Y. MUHAMMAD et al (2020), "Design of Fractional Swarm Intelligent Computing With Entropy Evolution for Optimal Power Flow Problems", IEEEE Access 8: 111401-111419.

• Section 1.2, second paragraph: when the authors refer to exploration vs. development, the correct terms should be exploitation vs. exploration. At least use "local exploration" to refer to exploitation.

• Section 2.2, Table 1: What "i", "j" and "N" represent is not correctly specified in the first line.

The sentence "There are other extended methods of this method, such as calculating the radius of a certain particle around the average position and calculating the average radius, which will not be detailed here" is really difficult to understand.

Equation 7 is not understood since the numerator (ki) was not defined. Correct: Chen -> Chen et al.

Rewrite the sentence "For the entropy measure, it is important to establish a model of discrete measurements, the swarm intelligence algorithm, for a large number of measurements, such as population location and population fitness".

Last paragraph: This last paragraph (Section 2.2) is incomprehensible. The method/idea should be more clearly described to the reader.

Typo: Oorunda -> Olorunda

• Section 2.3, check the final sentence in 4th paragraph : "a simple classification by fitness is rigorous" -> is not rigorous ?

• Section 2.4
Xu and Cui -> Xu et al.

Second paragraph, sentence "? intervals (? is the total number of particles)". However, in table 1, "n" represents the number of dimensions, whereas "M" represents the number of intervals. This is totally confusing for the reader.

The numerator of Equation 8 (Km,k) should be defined.

Fifth paragraph, sentence "Compared with the previous entropy method": What method?

• Section 2.5. Rewrite the second paragraph to clearly indicate each method to be used.  It is not clear what the authors refer to as "the average standard deviation of the four methods of diversity". It is unclear which are the two different entropy methods the authors use. Include the references in all methods.

• In experiment in Section 2.5.1, the authors do not specify the setup used for Edim (number of intervals M).

•  Section 2.5.3, rewrite the sentence "As the iteration times begin to approach zero, and when the late algorithm flocks to near zero", because I do not understand what the authors mean.

The PSO setup should be better detailed, including all the parameters related to the PSO moves of the particles. A note can be added indicating that PSO is described in the next section.

I do not understand the values in Table 2. The caption of Table 2 must be more descriptive. The values correspond to the Spearman correlation coefficient, but it is unclear what are the two variables whose correlation is been measured in each column.

• Section 3.1, I do not understand why the authors stablished that "The selection strategy in the DE algorithm is usually tournament selection". The DE selection operator is simple and is described in equation 15 (there are two Equations with number "14"), and not related with "tournament".

A table with the definitions of the test functions could be included. At least, a reference or link to the test set should be included. 

• Section 3.2: The authors indicate that "For this experiment, we selected the CEC17 single peak and the measure of the multi-peak value function". It is not clear what the authors mean by this sentence (and by the "measure"), since they used the 30 test functions. There are no bold values in table 3 (as the text specifies). The caption of the table should be more descriptive.

In the sentence "The experimental period was for 10 d" I suppose that the authors refer to the dimensionality considered. Rewrite the sentence to make it clear.

In the experiment in this section, the detailed setup of the three algorithms considered must be specified, in addition to the number of fitness evaluations. How were the parameters of each algorithm selected or tuned?

If an algorithm has parameters that perform more exploitation, the fitness would probably be worse (compared to the others) and the diversity would be lower. A correlation between these values (best fitness and final diversity) could be analyzed, but I do not see the usefulness of the analysis performed in Section 3.2, beyond comparing three algorithms under specific setups (not detailed to the reader).

The final sentence states that "A lower diversity slowly achieves convergence and maintains a balance between exploration and development. This may be a feasible improvement". I do not understand this statement, as low diversity means more exploitation and probably a fast convergence towards a local minimum, and it does not maintain a balance between "exploration and development".

• Section 3.3: The sentence "If a particle ?∈[1,2,…?] is in the largest set of dimensions ?∈ [1,2,…,?]," is unclear.

Algorithm 1: How is the value of the "expected entropy" set and in each iteration?

In the sentence "Setting a guidance algorithm to update the diversity of the benchmark is important", what is the "diversity of the benchmark"? I do not know what the authors mean by "the linear diversity guidelines". This paragraph must be completely rewritten.

Figure 6: The caption should be more descriptive, and the font of the text larger.

The second part of the last paragraph of Section 3.3 must be rewritten, it is incomprehensible.

• Section 4. The setup of the algorithms must be detailed. The results of the algorithms with and without the basic diversity control based on the dimension entropy, should be compared with other algorithmic variants that use diversity measures to adjust, for example, the defining parameters. For instance, a fair comparison should be the authors' mechanism incorporated in DE with respect to a DE version of self-adaptive parameters such as:
C. Laizhong et al. (2018), "Adaptive multiple-elites-guided composite differential evolution algorithm with a shift mechanism", Information Sciences 422: 122-143.
